# Exploring individual variation in associative learning abilities through an operant conditioning task in wild baboons

Claudia Martina[1,2]*, Guy Cowlishaw[2], Alecia J. Carter[1,2,3¤]*

**1** Department of Anthropology, University College London, London, England, United Kingdom, **2** The Institute of Zoology, Zoological Society of London, London, England, United Kingdom, **3** ISEM, Université de Montpellier, Montpellier, France

¤ Current address: Department of Anthropology, University College London, London, England, United Kingdom
* claudia.martinala@gmail.com (CM); alecia.carter@ucl.ac.uk (AC)

## Abstract

Cognitive abilities underpin many of the behavioural decisions of animals. However, we still have very little understanding of how and why cognitive abilities vary between individuals of the same species in wild populations. In this study, we assessed the associative learning abilities of wild chacma baboons (*Papio ursinus*) across two troops in Namibia with a simple operant conditioning task. We evaluated the ability of individuals to correctly associate a particular colour of corn kernels with a distasteful flavour through repeated presentations of two small piles of corn dyed different colours, one of which had been treated with a non-toxic bitter substance. We also assessed whether individual variation in learning ability was associated with particular phenotypic traits (sex, social rank and neophilia) and states (age and prior vigilance). We found no evidence of learning the association either within each trial or across trials, nor any variation based on individuals' phenotypes. This appeared to be due to a high tolerance for bitter foods leading to similar acceptance of both palatable and unpalatable kernels. Earlier avoidance of the bitter kernels during pilot trials suggests this higher tolerance may have been largely driven by a drought during the experiments. Overall, our findings highlight the potential influence of current environmental challenges associated with conducting cognitive tests of animals in the wild.

## Introduction

Learning results from past experiences which allow animals to adjust their behaviour accordingly [1]. Associative learning—a cognitive process that involves an association between stimuli and reinforcements—is key to many facets of animal behaviour [2], including fitness-related aspects such as foraging behaviour [cue preferences: 3,4; spatial memory: 5] and reproductive success [mate appeasement: 6; mate availability: 7; mating success: 8]. The costs and benefits associated to cues (i.e. highly negative or rewarding outcomes), will ultimately affect fitness, as they will determine the speed and strength with which novel associations are made

**Data Availability Statement:** All data files are available on Figshare: 10.6084/m9.figshare.9785867.

**Funding:** C.M. received funding from the following sources: 1. Consejo Nacional de Ciencia y

Tecnología (CONACyT, Mexico: http://www.conacyt.gob.mx/) 2. Fondo para el Desarrollo de Recursos Humanos (FIDERH, Mexico: https://www.fiderh.org.mx/) 3. School of Biological Sciences, University of Cambridge (Cambridge, UK: https://www.bio.cam.ac.uk/) 4 Hitchcock Fund, University of Cambridge (Cambridge, UK) The funders had no role in study, design, data collection and analysis, decision to publish, or preparation of the manuscript.

[9,10]. While ultimately, differences in associative learning abilities between species are likely to reflect adaptations [e.g. 11–13; but see: 14,15], inter-specific differences are also likely to reflect genotype [16,17] or epigenetic changes dependent on developmental trajectory and the environment experienced during their lifetimes [18,19].

One of the most commonly studied types of associative learning is operant conditioning, where learning is reinforced by the individual's own behaviour [20]. In his original work on operant learning, Skinner [21] trained pigeons and mice to press a lever to obtain a food reward until the animals pressed the lever continuously even in the absence of food. However, research on operant conditioning, as in other areas of animal cognition, has commonly relied on devices and protocols that restrict studies to captive conditions. While captive conditions offer a controlled environment in which to test animals [22,23], they may also limit the generalisability of the findings to cognition in the wild, for three reasons. First, captive research relies heavily on the training of animals for long periods of time [24–26], e.g., Okanoya et al. [26] demonstrated that captive caviomorph rodent degus (*Octodon degus*) are capable of tool use, but only after a training period of 2,500 trials. As such, conclusions obtained from captive studies may not be entirely representative of natural processes, particularly because opportunities to develop a given ability may be quite different for captive and wild animals. Second, captive animals often develop different behaviours to their wild conspecifics as a result of acquired experience through exposure to human-made objects or enforced proximity with conspecifics [27], e.g., captive animals often demonstrate greater diversity in tool use [27,28]. Conversely, the failure of captive animals to solve tasks that they would never encounter in the wild may equally distort our estimate of that species' cognitive abilities. This makes it difficult to predict how well the performance of captive subjects represents the abilities of wild conspecifics. Lastly, findings in captivity are likely to be influenced by the highly controlled conditions experienced by the study subjects [29], where the environmental and social aspects of behaviours that occur in natural circumstances, and that influence overall cognitive processing, are often ignored [30].

In this study, we investigated individual variation in the associative learning abilities of wild chacma baboons (*Papio ursinus*) by presenting individuals with an operant conditioning task that required them to associate colour with taste. Baboons, like humans, have trichromatic colour vision [i.e. they discriminate hues along the visible colour spectrum: 31], a trait predicted to have evolved out of the need to find ripe fruit amongst foliage [32]. The task presented here reflects a biologically relevant design as baboons may use colour changes in plant foods to help assess palatability [e.g., as fruits ripen: 33] and builds on previous studies of animal learning abilities that use colour cues during foraging [e.g. common bumblebees, *Bombus terrestris*: 3; pipevine swallowtails, *Battus philenor*: 34; greenfinches, *Carduelis chloris*: 35]. We tested for evidence of learning, with our null hypothesis being that the baboons would not learn the association between the colour (red or green) and palatability (palatable or bitter) of two food choices across five presentations (trials). We tested three possible mutually-exclusive processes about how the baboons could learn the association: we hypothesised that individuals would *either* rapidly learn the association between the colour and taste of two food choices during the first trial, after which individuals would choose only the food associated with the palatable colour in subsequent trials (learning process 1); *or* re-learn the association in each trial as independent events (failing to remember the association between trials), sampling both colours in each trial before selecting the palatable food (learning process 2); *or* gradually learn the association across trials, improving after each trial until they either largely/completely avoided the distasteful food, or preferred to consume the palatable option before the unpalatable one (learning process 3). In addition, for each of these three possible learning processes, we tested

five hypotheses regarding the source of individual variation in the learnt association according to five different phenotypic traits/states, as described below.

## Individual variation in associative learning ability: Hypotheses and tests

We tested three phenotypic traits (sex, social rank, neophilia) and two states (age, prior vigilance) that might explain individual differences in learning ability:

**Sex.** Sex differences have been recorded in a variety of cognitive abilities, including spatial cognition [36,37], innovation [38], and learning [39]. Furthermore, cognitive ability may be under sexual selection. Females may benefit from choosing males who, as a result of enhanced cognitive skills, are able to acquire more resources [40]; e.g., a male's success in a novel foraging task can correlate with his song complexity, a sexually selected trait females use to choose mating partners [zebra finches, *Taeniopygia guttata*: 41]. In this study, we predicted that female baboons would be more successful in a given task because, during gestation and lactation, females need to increase nutrient consumption while miminising exposure to plant secondary compounds [42,43], potentially selecting for more discriminative associative learning abilities where foods are concerned.

**Age.** Success or failure in cognitive tasks is often attributed to individual age [8,44]. For this task, we predicted that adult baboons would outperform juveniles. Although juvenile baboons have higher levels of exploratory behaviour than their older conspecifics [45,46], we might expect any associated advantage in cognitive testing to disappear once this has been accounted for (by controlling for individual neophilia, see below), such that adults outperform juveniles because of their greater experience identifying changes in food items with regards to their colour and taste.

**Social rank.** Cognitive performance may vary depending on the social rank of individuals [38,47,48]. Based on previous findings, we predicted two possible outcomes. On the one hand, dominants might outperform subordinates, for two reasons: because (*i*) they could have greater access to key resources (such as more nutritious foods in early life) that allow for better development and maintenance of cognitive abilities; and/or (*ii*) they are unlikely to be displaced and consequently can afford more time to solve a cognitive challenge [49,50]. On the other hand, subordinates might outperform dominants because low social status has promoted the development of cognitive abilities to circumvent traditional competition with dominants [51,52], for example, subordinate baboons are known to run ahead to access resources before the dominant arrives [53].

**Neophilia.** Individuals may differ in their reaction to novel situations: some animals avoid novelty, while others are attracted to it. Although both responses are sometimes considered opposite ends of the same continuum, they are independent of one another [54]. Neophobia, an aversion to novel stimuli [45], usually impedes cognitive performance [55] such that neophobic animals are less likely to fully engage with novel situations [48], whereas neophilia, an attraction towards novel stimuli [45,54], is associated with greater innovation and successful problem-solving as individuals are more likely to explore a novel situation [56,57]. Based on these previous findings, we predicted that more neophilic baboons would outperform less neophilic conspecifics.

**Vigilance.** Vigilance is defined as a state of frequent alertness that increases the likelihood of identifying possible changes or danger in individuals' immediate environment [58]. Vigilance often requires individuals to devote time and attention away from their current activity. When engaged in a novel task, this may compromise learning and reduce future task performance [40]. We predicted that baboons who were more vigilant during earlier experiences with a task, for instance because they perceive a higher risk of attack from conspecifics and/or

predators [e.g. social monitoring: 59; predator detection: 60], would perform less well the next time they attempted the task.

## Materials and methods

Fieldwork was carried out over a 6-month field season (April-September 2015) on two fully-habituated troops of chacma baboons, ranging in size from 43 (L troop) to 44 (J troop) individuals over four years of age (S1 Appendix) at Tsaobis Nature Park (15˚ 45'E, 22˚ 23'S) on the edge of the Namib Desert, Namibia. All these individuals were individually identifiable. Observers accompanied both troops on foot from dawn to dusk and used Cybertracker software (www.cybertracker.org) on smart phones (Samsung Galaxy S4) to record dominance and social interactions *ad libitum*. Our five individual traits/states were measured as follows:

i. *sex* was assigned by physical appearance (baboons are sexually dimorphic and have external genitalia);

ii. *age* was scored as juvenile or adult, where adulthood was defined by menarche in females and canine development in males. Generally, females become adults at 4–5 years of age while males become adult at approx. 9 years of age [61];

iii. *social rank* was determined using a dyadic winner-loser matrix in which all displacements, supplants, attacks, chases and threats observed between two individuals were recorded. Dominance hierarchies were calculated from the matrix using Matman software (Noldus Information Technology). In both troops, the hierarchy was strongly linear (Landau's corrected linearity index: $h'_J = 0.34$, $h'_L = 0.41$, $n_J = 946$, $n_L = 861$, $p < 0.001$ in both cases). Individuals' ranks were subsequently expressed relatively to control for differences in group size, ranging from 0 (lowest rank) to 1 (highest rank), using the formula $1-[(1-r)/(1-n)]$, where $r$ is the absolute rank of each individual and $n$ is the total group size [62];

iv. *neophilia* was measured using an established experimental approach in which individuals' responses to a novel food (an eighth of an apple dyed blue) were assessed using the time spent inspecting the item (s, median = 4; range = 1–120). All trials were conducted by AJC and have been shown to be repeatable in this population [$r = 0.26$, $P = 0.02$: 58]. For further details, see: [63]; and,

v. *vigilance* was defined as the number of times an individual lifted its head to scan its surroundings during the experimental trials and expressed as a rate per trial (the number of vigilance events observed divided by the total time of the trial (s)). We evaluated whether vigilance behaviour in the preceding trial affected individuals' responses in the current trial (e.g., the vigilance rate of trial 1 was used in trial 2).

### Experimental procedure

Individuals' associative learning abilities were evaluated with a task in which an association between the colour (red/green) and palatability (bitter/normal) of two piles of corn kernels had to be learned. Across Southern Africa, chacma baboons are notorious crop raiders of maize fields [64], which is a highly desirable and nutritious food. A simple pilot study was conducted by CM prior to this experiment, in Dec 2014 –Jan 2015, to observe individuals' responses to corn kernels soaked in a non-toxic concentrated bitter solution containing denatonium benzoate ('Avert', Kyron Laboratories Pty Ltd). During this pilot, two piles, one palatable and one unpalatable, of approx. 20 uncoloured kernels each, were presented on a single occasion to a random sample of 25 individuals (14 females: 13 adults, 1 juvenile); 11 males: 6

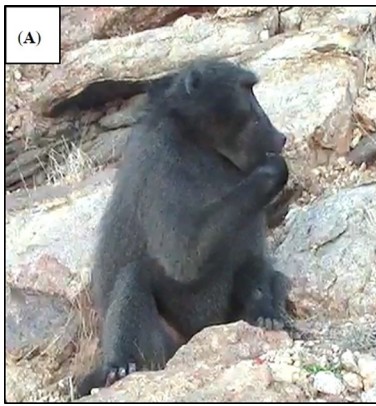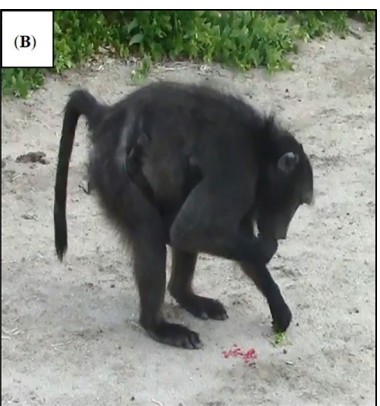

**Fig 1. Images of the baboons after being presented with both piles of coloured corn kernels.** Caption: Shown are (A) an adult male sitting down whilst eating the corn kernels, presented on a rocky surface and (B) an adult female bending over to eat the corn kernels, presented on sand.

adults, 5 juveniles). Individuals were deemed to be responsive to the bitter solution if they left kernels uneaten after tasting the bitter kernels. Baboons left kernels on 16 out of 25 occasions (64%) and spent a median of 12 s (range: 0–64 s) exploring the kernels (i.e. sniffing, rubbing kernels against forearm fur and/or repeatedly biting and spitting out the kernels). Such exploratory behaviour occurred in 21 out of 25 presentations (84%).

In the main experiment, a representative subset of 38 individuals was tested across the two troops. This involved 14 adult females, 3 juvenile females, 6 adult males, and 14 juvenile males, in each case, comprising 38–66% of the identifiable individuals in that age-sex class in our study population (S2 and S3 Appendices provide a breakdown of the number of baboons tested according to their sex-age class and their dominance and neophilia scores respectively). To keep a balanced sample it was necessary to include some individuals from our pilot study (n = 14); however, we considered it unlikely that these individuals' responses to the task would be any different, as all the baboons had prior experience with corn kernels [see: 65] and none had previously participated in a task involving colour cues indicating variation in palatability. Individuals were presented with two equal amounts of dried maize kernels (approx. 20 kernels each) of different colour and palatability, and their speed of learning this association was assessed over 3–5 presentations (median: 5 presentations). Corn kernels were initially soaked overnight in either a red or green edible food colourant (Moir's Food Dye); on the following night, one of these colours was soaked again in the bitter solution. For a similar methodology involving vervet monkeys, *Chlorocebus aethiops*, see [66]. Each troop was presented with a different unpalatable colour (green in J troop, red in L troop). All trials were conducted by CM and an assistant.

To avoid test subjects being displaced by dominant animals, or an audience learning socially by observing others, presentations were made to individuals when out of sight of conspecifics. CM and her assistant moved ahead of the foraging individual and waited until it was out of sight of others, at which point the assistant, who was positioned to record the trial a few meters ahead, indicated that the trial could start. CM then placed the two piles of corn on the ground ahead of the baboon while it was looking away. Each pile was approximately 10 cm in diameter and placed on the ground 10 cm apart from each other in a randomised left/right position to avoid any left/right preferences (Fig 1). Because trials could still be interrupted subsequently by other troop members, the same colour/palatability combination was used for all members of the same troop. All individuals received five "test" trials, each separated by

intervals of three days, i.e., after the first trial (day 0), individuals were tested on days 3, 6, 9 and 12. If it was not possible to test particular individuals on the assigned day, they were tested the next possible day (re-test interval (days): mean 3.7; median 3.0). Individuals who were tested fewer than four times ($n = 1$) were not considered for this analysis, with the exception of those tested in the month of May ($n = 9$: 2 adult females, 3 adult males, 2 juvenile females, 2 juvenile males), who could only be tested in three trials due to logistical complications. All tests were conducted between sunrise (0616–0632 during the testing period) and 1000 (mean testing time: 0745) to control for motivation, as individuals are more likely to have similar levels of hunger earlier in the day. A trial was considered as finished when the individual being tested consumed both piles of kernels in their entirety, walked a minimum of 2 m away from uneaten kernels, or were interrupted by conspecifics whilst they were at the task. We did not test any individual who interrupted a trial and ate from either of the corn piles to avoid the confounding effects of previous experience. We took particular care that those animals that were being tested did not observe or interrupt any of their conspecifics' evaluation until after their set of trials had finished. All experiments were filmed (Canon Vixia HF R300) to facilitate data extraction.

With the exception of three trials for which we were were entirely unable to extract data due to camera malfunction, the following data were obtained from the videos for each trial: (1) the colour of every kernel consumed; (2) the time spent eating each pile of kernels; (3) how many kernels were left (if any) from each pile; (4) the frequency of vigilance, measured as the number of times the individual scanned its surroundings by either lifting their head or noticeably moving their eyesight from the task; and (5) the total time dedicated to the task. While obtaining data from each video, we identified three potential sources of ambiguity, which we defined and addressed as follows: (1) when individuals' bodies blocked the task out of sight we noted the arm movement to control for the number of kernel being consumed, and coded those choices as missing values; (2) when a trial was cut short due to camera malfunction, individuals' first choice was noted and, depending on whether the malfunction occurred immediately at the start of the trial (i.e. before the 10 first kernels were consumed), that trial was not considered for further analysis; and (3) when individuals were joined by their kin (e.g. infants) during a trial, we took into account only those kernels chosen and consumed by the target animal, subtracting from the final amount the kernels consumed by their kin (if any). If the target individual moved away from the task immediately after being joined by their kin, we considered that trial as completed.

## Statistical analysis

All analyses were conducted in R (version 3.2.3, 2015). To test each of our proposed learning processes and their relationships to individual phenotype, we evaluated task performance in three ways which corresponded to the proposed learning processes, respectively. First, using kernel choice (binomial, Palatable, 1; Unpalatable, 0), we investigated every choice of kernel (1–40 kernels) in trial 1 to evaluate whether individuals were capable of learning the association rapidly, within a single presentation, after which they consistently avoided the unpalatable option in subsequent trials (the latter is tested in the subsequent models for trials 2–5) (learning process 1). Second, in a similar manner, we investigated every kernel choice (1–40 kernels) for trials 2–5, using separate models in each case, to test whether learning occurs independently in each presentation (learning process 2). Third, we used the proportion of correct kernels in the first 20 kernels eaten in trials 2–5 (numeric, 0–1) to test whether learning occurred gradually across trials (trial 1 was excluded as across-trial learning would only be evident in subsequent presentations, see below) (learning process 3).

**Table 1. Feeding patterns for the two piles of kernels in each trial.** The number of individuals and the feeding pattern adopted in each trial. The feeding patterns included individuals who: (*i*) switched between both piles presented within the first 20 kernels; (*ii*) "bulk" fed eating the palatable pile of kernels in its entirety before switching to the unpalatable one; and (*iii*) "bulk" fed eating the unpalatable pile of kernels in its entirety before switching to the palatable one. Also reported are the number of trials that were interrupted before individuals could sample both piles of kernels.

| Trial | Switch between piles within first 20 kernels | "Bulk feeding": palatable to unpalatable | "Bulk feeding": unpalatable to palatable | Interruptions |
|---|---|---|---|---|
| **1** | 13 | 13 | 5 | 3 |
| **2** | 15 | 10 | 7 | 5 |
| **3** | 14 | 11 | 6 | 7 |
| **4** | 7 | 9 | 9 | 3 |
| **5** | 12 | 4 | 2 | 4 |

To be able to learn the association between colour and palatability, the test subjects had to taste both types of kernel. We predicted that the baboons would have this opportunity by sampling both options at the beginning of the trial. However, this was often not the case, as the animals frequently "bulk" fed, eating one pile of corn entirely before switching to the next pile (Table 1). Consequently, we limited our analyses of within-trial learning (learning processes 1, 2) to those individuals that ate from both piles within the first 20 kernels (Table 1, first data column), i.e. all cases where individuals did not bulk-feed. However, for our analysis of across-trial learning (learning process 3), we used the proportion of correct kernels within the first 20 kernels in trials 2–5, as we assumed that even when individuals bulk-fed they would still acquire information about both piles of kernels by the end of trial 1 (provided they fed from both, which they did), which could then be applied in subsequent presentations.

We used generalised linear mixed-effects models (GLMMs) [package "lme4": 67] with a logit link function to account for binomial error structure to assess the effects of phenotype on task performance. Individual identity was included as a random effect in all models. To facilitate convergence, quantitative predictor variables were z-transformed to have a mean of zero and a standard deviation of 1. Preliminary analyses showed no co-variances where correlation was >0.70 between any of the fixed effects (S4 Appendix). Nevertheless, we tested each model for multicollinearity using variance inflated factors (VIFs) [package "usdm": 68]. As some of the fixed effects had a VIF of >2.0, we did a stepwise selection from the main model until all remaining variables had VIFs <2.0. To avoid overparameterisation, backwards elimination of non-significant terms was used, until a minimal model was obtained after which eliminated variables were then added back to the final model to check they remained non-significant. We describe each of the models in turn below.

**Learning process 1: Rapid learning in trial 1.** The analysis evaluating kernel choices within the first trial consisted of a model that addressed our questions about (a) how individuals learnt and (b) individual characteristics associated with variation in learning. As the response variable, the model ($M_{T1}$) included every kernel choice made (Palatable, 1; Unpalatable, 0) in this trial. To test for learning, we included the kernel number as a fixed effect. We predicted that learning would be demonstrated by a positive association between kernel number and the probability of consuming a palatable kernel. Additionally, we included interactions between kernel number and the sex, age, social rank and the neophilia level of individuals. A significant interaction with any of these variables would provide evidence of phenotypic trait/state-dependent learning differences. Three further fixed effects were also included: (1) individuals' first choice of kernel in that trial (Palatable, P; Unpalatable, U), to control for those individuals that may have found it more difficult to detect a palatable kernel when tasting the bitter kernels first; (2) troop identity, to control for the possibility that baboons have an innate preference for a particular food colour; and (3) the probability of randomly selecting a palatable kernel at each choice, to account for the change in the proportion of palatable choices

available as the trial progressed. This final variable was calculated as the proportion of remaining kernels that were the "palatable" choice such that at the start of the trial this proportion was 0.50 (20 of 40 kernels), and was subsequently updated with each choice that was made until no choice was available (i.e. one pile had been consumed in its entirety). When individuals consumed a pile in its entirety, they were no longer able to choose between the two options; as such, we did not consider individuals' choices after all the kernels of one pile were eaten. When trials were interrupted or individuals left kernels uneaten, a missing value was assigned to the remaining choices that were no longer possible to make. Evidence for the first learning process, that individuals learnt the association in the first trial and remembered the association, would involve not only a positive relationship between kernel choice and number in this model but also consistently palatable choices in the subsequent trials, which are analysed in the next set of models (see below).

**Learning process 2: Repeated rapid learning in trials 2–5.**    To test whether individuals re-learnt the association in each trial, we fitted four further models following the same model structure outlined in $M_{T1}$ above for each of the subsequent trials 2–5 ($M_{T2}$, $M_{T3}$, $M_{T4}$ & $M_{T5}$).

**Learning process 3: Gradual learning across trials.**    Gradual learning may be identified by an increasing number of palatable kernels eaten across trials, until only the palatable kernels are eaten at the start of a trial. We therefore analysed the proportion of palatable kernels eaten of the first 20 kernels in trials 2–5, as this amounts to the quantity of one pile of kernels. Trial 1 was not analysed in this sample, as this was the initial learning opportunity. The fixed effects in this model (model $M_{T2-5}$) comprised trial number, to assess whether there was evidence for gradual learning; individual traits/states and their interactions with trial number, to assess whether gradual learning was predicted by individuals' phenotypic traits/states; and two of the same additional fixed effects as in the preceding models, i.e. an individual's first choice of kernels (in the first trial) and troop membership. In this model, we also evaluated past vigilance behaviour as a predictor, to test whether learning was negatively affected by an individual's attention being diverted from the task. This required the inclusion of two additional fixed effects in the model, the number of vigilance instances observed in the previous trial (i.e. the vigilant behaviour of trial 1 was used as a predictor of trial 2) and the total time of each corresponding trial.

**Interpretation of within-trial findings.**    Throughout the course of the experiment, the baboons adopted two unanticipated behaviours that complicated the interpretation of the within-trial learning results: bulk-feeding and consumption of the unpalatable kernels. Bulk-feeding resulted not only in reduced sampling of the available options at the start of the trials but also in reduced switching between options within trials, even if individuals had sampled both options within the first half of the trials. This affected the probability of choosing a palatable kernel as the trial progressed as one option was depleted continuously for an extended number of choices, even if it was the unpalatable choice. In addition, consumption of the unpalatable kernels meant that learning could be masked in the analyses. This is because individuals who chose the correct kernels first and then switched to the unpalatable kernels would, counterintuitively, show a *negative* probability of choosing the palatable kernels as the trial progressed, even if they had learnt the colour association and as a result were choosing the palatable option first. Because these unanticipated behaviours complicated the interpretation of the results in ways that were difficult to predict intuitively, we ran simple *post-hoc* scenarios of the possible outcomes (i.e. the observed relationships between kernel choice and both kernel number and the probability of randomly choosing the palatable kernel) that included the options of bulk-feeding and consumption of the unpalatable kernels to determine how we might still identify learning under these circumstances.

We plotted five scenarios of the different within-trial processes, two that assumed within-trial learning and three that assumed no learning: (1) fast learning at the start of the trial; (2) slow learning throughout the trial; (3) no learning (without bulk feeding); (4) bulk-feeding on the palatable kernels at the start of the trial; and (5) bulk-feeding on the unpalatable kernels at the start of the trial. Each scenario generated a visual plot against which our observed data were compared. In the first case, individuals alternately sampled two of each kernel to learn the association and, starting from the fifth kernel, then ate the remaining palatable kernels before switching to the unpalatable kernels. In the second case, individuals started in the same manner as for fast learning, but progressively ate more of the palatable kernels while still inter-mittently sampling 1–3 unpalatable kernels until no correct kernels remained and the unpalat-able kernels were then consumed. In the third case, the choice was random. In the fourth and fifth cases, individual sampled one unpalatable or palatable kernel, respectively, first, before bulk-feeding on the other option. All scenarios were based on observations of feeding patterns in those trials in which individuals switched between piles. Scenarios were not probabilistic and as such there is only one outcome for each scenario as described. For each scenario, we plotted the proportion of palatable kernels eaten relative to (1) the kernel number and (2) the probability of choosing the palatable kernel given the proportion of palatable kernels that remained (Fig 2). We used these plots to generate expectations of what the data would look like under each scenario against which we could compare our observed results for both trials 1 (learning process 1) and trials 2, 3, 4, and 5 (learning process 2) (Table 2).

### Ethics statement

Our research and protocols were assessed and approved by the Ethics Committee of the Zoo-logical Society of London (BPE 727) and approved by the Ministry of Environment and Tour-ism in Namibia (Research Permit 2009/2015). We contacted private and public landowners directly and were given full permission to work on their land throughout the field season.

### Results

We tested 38 individuals over 162 trials overall (mean number of presentations = 4.3; median = 5; range 3–5). In total, 43 trials (26%) were interrupted by displacements or sup-plants by more dominant animals, while 22 (13%) interruptions happened before animals could sample both piles of kernels. None of the individuals tested in each trial subset were interrupted before they had sampled both options. Trials lasted on average 15 s (range = 1–545). Compared to the pilot study where 64% of individuals left unpalatable kernels uneaten in the single presentation (see Experimental Procedure), in the current experiment only 16% of indi-viduals (6 of 38) left unpalatable kernels uneaten in the first presentation ($X^2$ = 10.38, $p$ = 0.001). Across all uninterrupted trials (119), the baboons consumed a median of 11 palat-able kernels in the first 20 kernels (range 0–20) and ate both piles of corn in their entirety in 79 (48%) uninterrupted trials. In the remaining trials, the median number of palatable kernels remaining was 15 (range 10–20), in comparison to a median number of 18.5 unpalatable ker-nels (range 8–20). These patterns suggest little discrimination between the palatable and unpalatable kernels. A similar pattern was seen at the individual level: among the 12 individu-als who completed five uninterrupted trials, on 36 out of 47 occasions (77%) all the corn was consumed irrespective of palatability. On average, animals had 4 instances of vigilant behav-iour (range = 0–18) per trial. Sniffing behaviour was observed in 21% of trials; however, pre-liminary tests showed no relation with the choice of kernels eaten in each trial, nor the proportion eaten across trials.

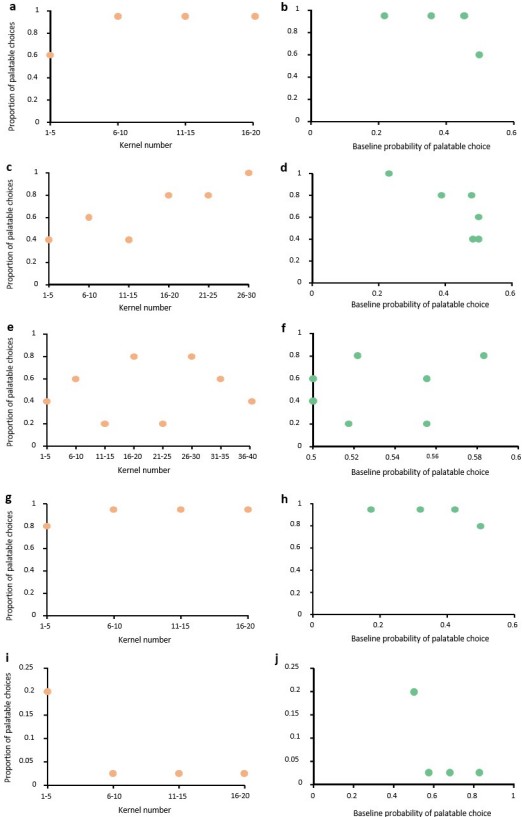

**Fig 2. Plots of the possible learning scenarios within each trial.** Caption: Shown are the relationships for the learning scenarios for: (a, b) fast learning; (c, d) slow learning; (e, f) no learning; (g, h) palatable bulk-feeding; and (i, j) unpalatable bulk-feeding. Each pair of plots shows the relationship between the proportion of correct kernels eaten and either the kernel number (left plot, in orange) or the probability of choosing a correct kernel (right plot, in green). Kernel number indicates the proportion of palatable kernels eaten in groups of five. The probability of randomly choosing the palatable kernels shown in each plot represents the median probability of palatable choices in groups of five. Note that the number of points is contingent upon how quickly the baboons completely consumed one of the piles, after which no choice was possible, and the scenario ended. Differences in the x and y-axis reflect the differences in the proportion of palatable kernels and probability based on the learning scenario proposed.

In our within-trial analyses, we found no consistent evidence of associative learning in either the first trial (rapid learning, process 1) or the subsequent four trials (repeated rapid learning, process 2), due to the lack of a consistent positive relationship between correct kernel choice and kernel number (Table 3). A comparison of our results with our within-trial learning scenarios suggests our findings are more consistent with individuals using a bulk-feeding pattern after sampling with no learning (Table 4). The patterns are not consistent between trials; however, our results show baboons favoured bulk-feeding from the unpalatable pile of kernels in three out of five trials; while in only one out of five they favoured bulk-feeding on the palatable kernels. Results in the last trial ($M_{T5}$) were inconsistent with any of the five learning scenarios tested. A plot of the raw data for kernel choice over time in each of the five trials (which does not control for changes in the availability of palatable kernels over time) is provided for illustrative purposes in Fig 3. Regarding evidence of across-trial learning (Process 3), this analysis only yielded a significant relationship between the proportion of palatable kernels eaten and troop membership ($M_{T2-5}$, Table 3, Fig 4), where individuals in L troop were more likely to eat a higher proportion of unpalatable kernels across trials than individuals in J troop.

There was no evidence of learning across trials, i.e. the proportion of palatable kernels eaten did not increase in later trials.

Lastly, we found little evidence of between-individual differences in task performance. There was only one model in which an effect of phenotype was detected: an interaction between neophilia and kernel number was found to influence kernel choice in trial 2 ($M_{T2}$, **Table 3**, Fig 5), where individuals of high and low neophilia differed in their relative consumption of palatable kernels in the middle of each trial.

## Discussion

We tested the associative learning abilities of individuals belonging to two groups of wild baboons with an operant conditioning task involving an association between the colour and taste of corn kernels (red/green, palatable/unpalatable) over five trials. We expected that all individuals would show an improvement in task performance as they learned the colour-taste association either within or across trials, and that certain phenotypes would show faster learning than others. However, we did not find support for any of our predictions. Overall, our results suggest that individuals bulk-fed without learning on either the palatable or unpalatable kernels in most trials. Additionally, we found limited evidence of individual differences in learning, as there was only a single significant interaction between kernel number and neophilia, where individuals with lower levels of neophilia were more likely to eat the correct kernels at the start of that trial. We also found that troop membership determined the likelihood of eating a higher proportion of correct kernels across trials.

An animal's fitness may depend on its ability to make rapid associations regarding novel foods [69], as animals must not only determine the safety of those foods but also whether they are nutritionally rewarding [70,71]. Previous studies with chacma baboons provided further support to our expectation that baboons would concisely and rapidly learn the taste-colour association. For example, captive baboons were able to solve complex learning tasks using an automated device [72], while wild baboons are capable of rapidly learning the location of valuable food items [e.g. 73,74]. Although we found no evidence for learning, it is possible that the baboons learnt there was a difference in taste between both piles of kernels (i.e. one was bitter while the other was not) but were still willing to eat the bitter kernels. This interpretation is supported by our observation that the baboons left slightly more bitter kernels uneaten. Generalist species such as baboons can adapt quite successfully to situations involving novel-flavoured foods [75]. This may be in part because they have relatively low gustatory sensitivity and can readily incorporate novel foods into their diet even when these are unpalatable to other species [76]. Captive studies indicate that tolerance to bitterness varies among primate species, particularly to compounds not found in nature, such as the bitter substance used here

**Table 2. Possible within-trial learning scenarios and simulated outcomes.** The proposed learning scenarios of the within-trial learning process and the expected directions of the effect for the fixed effects of kernel number and the probability of making the palatable choice.

| Proposed scenario | Predicted effect of kernel number on the response | Predicted effect of probability of randomly choosing the palatable choice on the response |
|---|---|---|
| **Fast learning** | Positive (weak) | Negative (weak) |
| **Slow learning** | Positive (strong) | Negative (strong) |
| **No learning** | None | None |
| **Bulk feed on palatable kernels** | Positive (very weak)/none | Negative (very weak)/none |
| **Bulk feed on unpalatable kernels** | Negative/none | Negative/none |

**Table 3. Predictors of performance in wild chacma baboons in an associative learning task with coloured corn.** The three learning processes are evaluated in turn (divided by the bold line). Each model is listed by name (see text for details) and the learning process being tested; the response variable; the number of observations and individuals; the deviance; and the fixed effects of the minimal models, with their effect sizes and standard errors (coefficient and its equivalent probability in parenthesis, S. E.), test statistic (t) and p-values. Significant results with values of $p < 0.05$ are highlighted in **bold**. [1]Reference category: J troop. [2]Reference category: Palatable kernels.

| Model | Learning Process | Response | Nobs/ Nind | Deviance | Term | Coefficient | S.E. | t | p |
|---|---|---|---|---|---|---|---|---|---|
| $M_{T1}$ | 1: Rapid Learning | Kernel choices in Trial 1 | 374 / 13 | 476.6 | Intercept | 1.02 (0.73) | 0.28 | 3.60 | |
| | | | | | Kernel Number | -0.02 (0.49) | 0.01 | -2.15 | **0.03** |
| $M_{T2}$ | 2: Repeated Rapid Learning | Kernel Choices in Trial 2 | 303 / 15 | 287.1 | Intercept | 2.33 (0.91) | 0.62 | 3.70 | |
| | | | | | Kernel Number | -0.03 (0.49) | 0.02 | -1.47 | 0.13 |
| | | | | | Neophilia | -0.65 (0.34) | 0.44 | -1.46 | 0.14 |
| | | | | | Troop: L[1] | -1.78 (0.14) | 0.79 | -2.25 | **0.02** |
| | | | | | Probability of correct choice | -0.69 (0.33) | 0.39 | -1.77 | 0.07 |
| | | | | | K. Number*Neophilia | 0.07 (0.51) | 0.02 | 3.24 | **0.001** |
| $M_{T3}$ | 2: Repeated Rapid Learning | Kernel Choices in Trial 3 | 327 / 14 | 332.9 | Intercept | -1.67 (0.15) | 0.65 | -2.55 | |
| | | | | | Kernel Number | 0.12 (0.52) | 0.02 | 5.87 | **<0.001** |
| | | | | | Troop: L[1] | 1.55 (0.82) | 0.85 | 1.82 | 0.06 |
| $M_{T4}$ | 2: Repeated Rapid Learning | Kernel Choices in Trial 4 | 134 / 7 | 141.7 | Intercept | 1.91 (0.87) | 0.51 | 3.70 | |
| | | | | | Probability of correct choice | -2.32 (0.08) | 0.67 | -3.46 | **<0.001** |
| $M_{T5}$ | 2: Repeated Rapid Learning | Kernel Choices in Trial 5 | 180 / 12 | 173.6 | Intercept | -4.98 (0.006) | 1.87 | -2.66 | |
| | | | | | Kernel Number | 0.71 (0.67) | 0.19 | 3.73 | **<0.001** |
| | | | | | First Choice: U[2] | 3.56 (0.97) | 1.79 | 1.98 | **0.04** |
| | | | | | Probability of correct choice | 12.57 (0.99) | 3.35 | -3.75 | **<0.001** |
| $M_{T2-5}$ | 3: Across-trial Learning | Proportion of correct kernels in Trials 2–5 | 111 / 38 | 509.8 | Intercept | 2.75 (0.93) | 0.84 | 3.26 | |
| | | | | | Troop: L | -2.37 (0.08) | 1.17 | -2.02 | **0.04** |

(denatonium benzoate) [77]. An additional aspect in this study is that it was conducted during a severe drought year, which likely further influenced the test subjects' willingness to accept the bitter food presented. Chacma baboons elsewhere in the Namib Desert have been reported to feed on unpalatable toxic plants that they would normally avoid, such as *Euphorbia avas-*

**Table 4. Observed learning scenarios within each trial.** Caption: The table shows the observed relationships in the GLMMs (from Table 3) and the inferred learning scenarios based on these relationships (from Table 2). Shown are: the trials and corresponding models; the direction of the observed main effect of kernel number on kernel choice; the direction of the observed main effect of probability to choose the palatable kernel and kernel choice; and the scenario that best fits the observed data based on the predictions generated from the learning scenarios. A question mark (?) indicates that the results obtained did not match any of the simulated learning scenarios. *The interpretation of this learning scenario is complicated due to the presence of a significant interaction in the minimum model. For the purposes of comparison with the simulation output, we consider the direction of the estimate of the main effects only.

| | Observed effect of kernel number on the response (from Table 3) | Observed effect of probability of a correct choice on the response (from Table 3) | Best fit scenario (from Table 2) |
|---|---|---|---|
| **Trial 1 ($M_{T1}$)** | Negative | None | Bulk feed on unpalatable kernels |
| **Trial 2 ($M_{T2}$)** | Negative | Negative | Bulk feed on unpalatable kernels* |
| **Trial 3 ($M_{T3}$)** | Positive | None | Bulk feed on palatable kernels |
| **Trial 4 ($M_{T4}$)** | None | Negative | Bulk feed on unpalatable kernels |
| **Trial 5 ($M_{T5}$)** | Positive | Positive | ? |

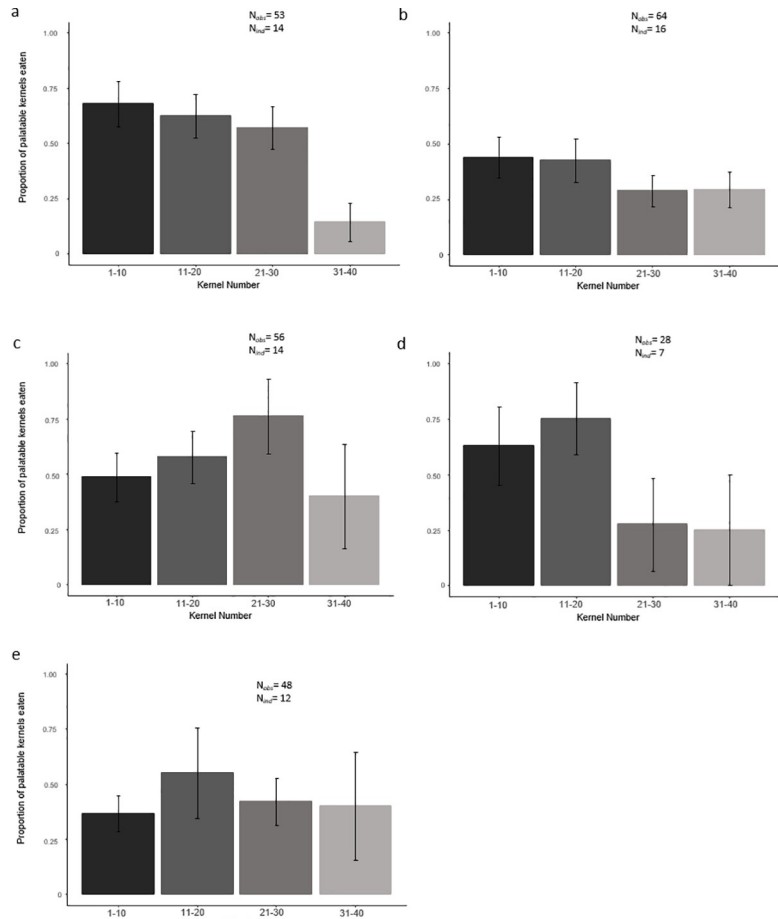

**Fig 3. Plot of the relationship between the proportions of correct kernels chosen and kernel number in trials 1–5.**
Caption: Bar-plot of the proportions (mean ± S.E.) of palatable kernels chosen in the first (1–10), second (11–20), third (21–30) and fourth ten (31–40) kernels consumed in (a) trial 1; (b) trial 2; (c) trial 3; (d) trial 4; and (e) trial 5. Each plot shows the raw data and therefore does not control for the changing availability of palatable kernels. The proportion of palatable kernels (0–1) in each plot was calculated as the number of palatable choices in every group of ten kernels divided by ten. If a trial was interrupted before consuming the fifth palatable kernel within a given set of ten, those data were omitted from the plot. However if the trial was interrupted when the minimum of five palatable kernels or more within a given set of ten had been consumed, the final number of palatable kernels within that set was divided by the total number of kernels in that set and included in the plot. Standard error bars are shown for each column in each plot.

*montana*, during drought conditions [78]. The observation that the baboons showed a stronger rejection of the bitter kernels during the pilot study, when environmental conditions were less severe than during the main study conducted five months later, supports this interpretation. Additionally, it is possible that the acceptance of bitterness was reinforced by the baboons' previous experience with kernels, which represented a nutritious and safe source of food.

Our analyses of learning across trials revealed that individuals from J troop, in which the palatable kernels were red, tended to eat a higher proportion of these kernels within their first 20 choices than their L troop conspecifics. Such a result may be indicative of a species' selective attention towards red food items. The ripening of fruit from green to red correlates with changes in their glucose levels, indicating their taste quality and nutritional value [31]. Moreover, while the normal diet of both troops includes large amounts of immature green pods,

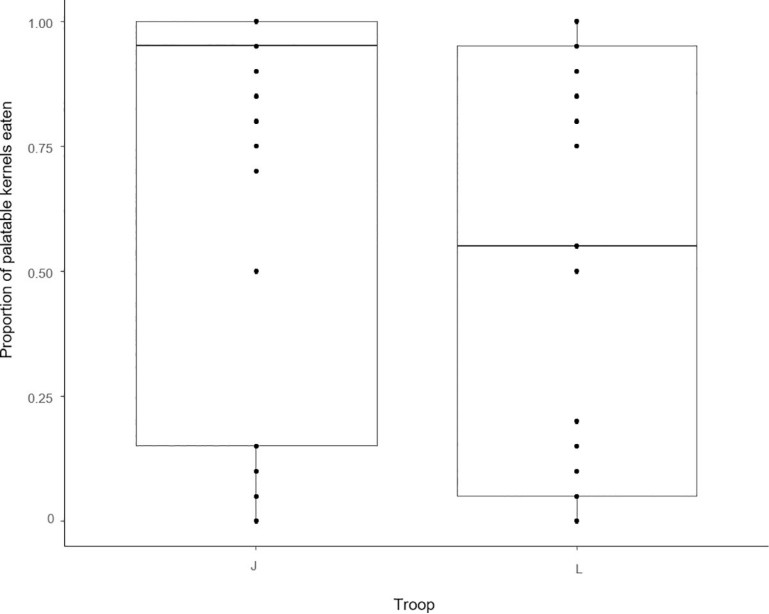

**Fig 4. The relationship between troop membership and the proportion of correct kernels eaten within the first 20 choices across trials 2–5.** Caption: Box-and-whisker plot of the proportions of palatable kernels chosen within the first 20 choices for each troop. Cases in which individuals who did not consume at least 20 kernels were not included in the sample analysed. The horizontal line in each box indicates the median, the box shows the lower (25%) and upper (75%) quartiles of the data, and the whiskers the minimum and maximum values. Black dots represent individual values.

leaves and stalks, individuals who were tested could have had a natural preference for the red kernels as this colour represents a key seasonal fruit in this environment: the ripe winter berries of *Salvadora persica*. Since we were avoiding any potential habituation or possible generalisation between colours, we did not test for a specific colour preference to exclude; however, as both colours used in this task represented food items frequently eaten, and thus, ecologically relevant to learn, it was not obvious at the time that the baboons would have a preference for either colour. The fact that this result was only inconsistently observed within trials suggests that the preference may be a relatively weak one (perhaps unsurprisingly, given foods of both colour occur in the natural diet), and was therefore best captured by sampling individual preferences over the larger range of choices analysed in the across-trials model.

Some unexpected patterns were observed in the within-trial analysis, in particular, the failure of the results from trial 5 to correspond with any of the simulated learning scenarios, and the interaction between neophilia and kernel number that was contrary to prediction in trial 2. In the first case, the characteristics of the subset may be responsible: an unusually high proportion of trials were interrupted in this final trial (7 out of 15 trials) and early and late interruptions (i.e. within the first 10 and 20 kernels respectively) were not considered in the learning scenarios. In the second case, we found that more neophilic and less neophilic individuals differed in their relative consumption of correct kernels in the middle of this trial. However, because this was only observed in a single trial, and after a prior presentation, it is difficult to interpret. Given the number of interactions we tested, and our decision not to apply Bonferroni corrections [79], the most parsimonious interpretation may be that this result represents a spurious relationship.

Our experience in developing and conducting this study reflects some of the challenges involved in devising cognitive tasks that efficiently assess cognitive abilities and suit the species under study, particularly in wild conditions [80,81]. The pilot tests we conducted saw a

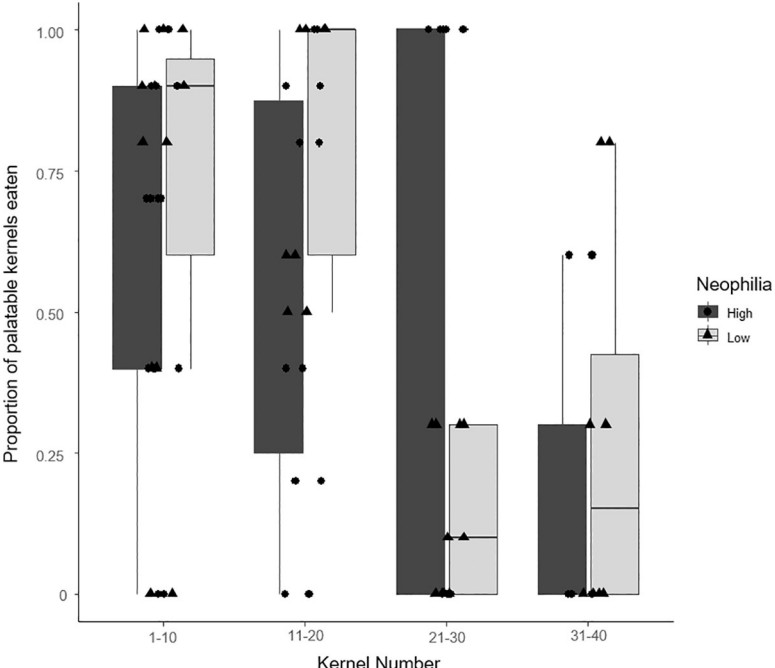

**Fig 5. The interaction between kernel number and neophilia on kernel choice in trial 2.** Caption: Plot of the proportion (mean ± S.E.) of palatable kernels chosen in the first (1–10), second (11–20), third (21–30) and fourth ten (31–40) kernels consumed in trial 2 and the level of neophilia of individuals (High/Low). The proportion of palatable kernels (0–1) was calculated as the number of palatable choices in every group of ten kernels divided by ten. If a trial was interrupted before consuming the fifth palatable kernel within a given set of ten, those data were omitted from the plot. However if the trial was interrupted when the minimum of five palatable kernels or more within a given set of ten had been consumed, the final number of palatable kernels within that set was divided by the total number of kernels in that set and included in the plot. This plot shows the raw data and therefore does not control for the changing availability of palatable kernels. The whiskers shown on each dot represent the minimum and maximum values. For visualization purposes, neophilia level was grouped evenly into two categories, high and low according to the median value. Black circles and triangles represent individual values for high and low neophilia respectively.

majority of individuals avoiding the maize after tasting the bitter kernels, suggesting they were sensitive to such a substance. However, it is possible that differences in the abundance of natural foods during the pilot study, which was conducted over a period of summer rains, and during the experiment, which was conducted over a period of winter drought, resulted in the latter differing substantially from what was observed in the former. Despite substantial literature detailing the influence of unfavourable environment on animals' cognitive abilities [e.g. 82,83] and cue use [e.g. 35], there is little empirical evidence on the phenotypic plasticity of labile traits related to cognition. Consequently, we have a poor understanding of cognitive variability in fluctuating, rather than predictable and constant, environments. Future studies could consider differences in learning behaviour relative to the resources available in the environment at different points in time. Additionally, the unanticipated performance of bulk feeding behaviour made this standard laboratory experiment difficult to interpret. Because the baboons bulk fed from one pile of corn at a time, few individuals sampled both colours of corn at the start of a trial. Because of this, we were forced to analyse a smaller sample, which reduced the statistical power to detect an effect or pattern. Ultimately, our study highlights the importance of using the right task to assess cognitive abilities, taking into account not only the natural behaviour of animals, but also their current environmental conditions to understand how abilities such as associative learning develop in a natural setting.

## Supporting information

**S1 Appendix. Demographic breakdown of all identifiable individuals in both study troops.**
(DOCX)

**S2 Appendix. The numbers of baboons tested according to sex, age and troop in the study.**
Shown are the total numbers of baboons tested of each age-sex class in each troop, including
the total. The percentage of the population that the sample represents is presented in brackets.
(DOCX)

**S3 Appendix. The numbers of baboons tested according to dominance rank and neophilia
level in the study.** Shown are the total numbers of baboons tested of each dominance rank
and neophilia level, including the total. The percentage of the population that the sample rep-
resents is presented in brackets. For the purposes of this table, dominance ranks were grouped
evenly into categories of "low-rank", "medium-rank" and "high-rank" according to tertiles;
while neophilia levels was grouped evenly into categories of "low-neophilia", "medium-neo-
philia" and "high-neophilia" according to tertiles.
(DOCX)

**S4 Appendix. Spearman rank correlation coefficients of the predictor variables used in the
GLMM models.** Shown are the Spearman correlation coefficients of the predictor variables
used in the GLMM models. Sample size is N = 38 individuals in all cases. Individual vigilance
and total time were calculated as the median across all trials (1–5). First choice refers to the
first choice between each pile of corn in the first trial (Palatable, P; Unpalatable, U).
(DOCX)

## Acknowledgments

We would like to thank all the Baboon Team 2015 for their incredible patience and persever-
ance chasing after baboons and CM, in particular: C. Rice, I. Knot, T. Argouges, M. Rizzutto,
G. Powell and I. Hirschler. We would like to additionally thank Rachel Kendall and Stuart
Semple for their comments on an earlier version of this manuscript; Stefan Fischer for his sug-
gestions and help with the statistical analyses; and Volker Sommer for his continuous support.
We thank the Ministry of Lands and Resettlement for permission to work at Tsaobis Nature
Park, the Gobabeb Training and Research Centre for affiliation, and the Ministry of Environ-
ment and Tourism for research permission in Namibia. We are also grateful to the Snyman
and Wittreich families for permission to work on their land. We thank the following organisa-
tions for providing the funds to conduct this research: the Consejo Nacional de Ciencia y Tec-
nología (CONACyT, Mexico), the Fondo para el Desarrollo de Recursos Humanos (FIDERH,
Mexico), the Cambridge Trust Fund, the Hitchcock Fund and the School of Biological Sciences
at the University of Cambridge. Lastly, we thank our two anonymous reviewers for their com-
ments on an earlier version of this manuscript. This paper is a publication of the ZSL Institute
of Zoology Tsaobis Baboon Project. ISEM contribution number 2020–062.

## Author Contributions

**Conceptualization:** Claudia Martina, Guy Cowlishaw, Alecia J. Carter.

**Formal analysis:** Claudia Martina.

**Funding acquisition:** Claudia Martina.

**Methodology:** Claudia Martina.

**Project administration:** Alecia J. Carter.

**Supervision:** Guy Cowlishaw, Alecia J. Carter.

**Writing – original draft:** Claudia Martina.

**Writing – review & editing:** Claudia Martina, Guy Cowlishaw, Alecia J. Carter.

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
