## [Decision Letter · Decision Letter 0]

25 Nov 2019

PONE-D-19-25653

Exploring individual variation in associative learning abilities through an operant conditioning task in wild baboons

PLOS ONE

Dear Dr. Carter,

Thank you for submitting your manuscript to PLOS ONE. After careful consideration, we feel that it has merit but does not fully meet PLOS ONE’s publication criteria as it currently stands. Therefore, we invite you to submit a revised version of the manuscript that addresses the points raised during the review process.

Unfortunately, Reviewer 1 was no longer available, and a new reviewer was invited.

Both reviewers found that your study is valuable and that performing cognitive studies in the wild is extremely challenging, and I share their view. However, whereas Reviewer 1 (previously Reviewer 2) was quite happy with your revision, the current Reviewer 2 asked for further work on the paper, before it can be considered acceptable. Please address the comments provided by both reviewers, along with my suggestions, listed below. 

We would appreciate receiving your revised manuscript by Jan 09 2020 11:59PM. To enhance the reproducibility of your results, we recommend that if applicable you deposit your laboratory protocols in protocols.io, where a protocol can be assigned its own identifier (DOI) such that it can be cited independently in the future. For instructions see: http://journals.plos.org/plosone/s/submission-guidelines#loc-laboratory-protocols

We look forward to receiving your revised manuscript.

Kind regards,

Elsa Addessi

Academic Editor

PLOS ONE

Journal Requirements:

3. Please include a copy of Table 2 which you refer to in your text on page 23.

Additional Editor Comments (if provided):

ll 39-40: please modify as follows: “current environmental challenges associated with conducting cognitive tests of animals in the wild”

Ll 494-495 “We also found that troop membership determined the likelihood of eating a higher proportion of correct kernels across trials.”: couldn’t it be done to color preference? Did you test color preference before administering the differently flavored kernels? Although I appreciated the answers you provided in reply to the comments of the previous reviewers on this issue, this aspect should be accounted for in the Discussion.

L 529 “and the interaction that was contrary to prediction in trial 2”: please remind the reader what is the interaction you are mentioning here

L 533: “that” repeated twice (typo)

Reviewers' comments:

Reviewer's Responses to Questions

**Comments to the Author**

1. Is the manuscript technically sound, and do the data support the conclusions?

Reviewer #1: Yes

Reviewer #2: Partly

2. Has the statistical analysis been performed appropriately and rigorously? 

Reviewer #1: Yes

Reviewer #2: No

3. Have the authors made all data underlying the findings in their manuscript fully available?

Reviewer #1: Yes

Reviewer #2: Yes

4. Is the manuscript presented in an intelligible fashion and written in standard English?

Reviewer #1: Yes

Reviewer #2: Yes

5. Review Comments to the Author

Reviewer #1: General comments

This study aimed to assess associative learning performance of individuals in two groups of wild baboons. The manuscript represents a significant amount of work and while it is unfortunate that the authors were not able to detect learning, I believe the trial was thoughtfully conducted. There remains a paucity of studies investigating cognition in wild populations (it is challenging!) and studies such as the current one are useful building blocks for future attempts. The authors have done a nice of addressing the concerns/ suggestions raised in the previous two reviews, as such I only have minor comments.

Minor comments:

1. L51 – As there are only a small number of studies that have looked at the fitness consequences of associative learning (adaptive value) in natural environments, it would be good to cite them all here. Also, as the authors correctly point out, differences in learning are likely to reflect adaptions. The only study currently cited reports a positive correlation between learning performance and fitness correlates, but two other studies report negative relationships:

Ashton B.J., Ridley A.R., Edwards E.K., Thornton A. 2018 Cognitive performance is linked to group size and affects fitness in Australian magpies. Nature 554 (7692), 364-367 (doi:10.1038/nature25503).

Madden JR, Langley EJG, Whiteside MA, Beardsworth CE, van Horik JO (2018) The quick are the dead: pheasants that are slow to reverse a learned association survive for longer in the wild. Philosophical Transactions of the Royal Society B: Biological Sciences 373

Evans LJ, Smith KE, Raine NE (2017) Fast learning in free-foraging bumble bees is negatively correlated with lifetime resource collection. Scientific Reports 7:496

2. L52 – This sentence should be modified to improve clarity. I suggest inserting ‘while ultimately differences in…’ removing also from ‘individuals within a species also differ…’ and including ‘genotype or epigentic changes due to the developmental trajectory environment experienced during their lifetime.

3. L85 – Incorrect references inserted, should be references ‘3’ (bumble bees) and ‘4’ (humingbirds). It doesn’t seem as though reference ‘5’ has been used at all – so it would pay to double check all references align as intended.

4. L86 – Rather than including a second reference for bumble bees can you reference a study that has looked at associative colour learning in foraging mammals? Or a least a different taxon e.g. butterflies, jumping spiders.

5. Discussion – it would be good to briefly comment on baboon cognition generally. What would you expect to see in terms of their associative learning performance, based on their performance in other cognition assays, conducted either with captive or wild individuals?

6. L552 - It would be good to reinforce that studying cognition in wild populations is generally very challenging by citing the below studies:

Huebner F, Fichtel C, Kappeler PM (2018) Linking cognition with fitness in a wild primate: fitness correlates of problem-solving performance and spatial learning ability. Philosophical Transactions of the Royal Society B: Biological Sciences 373

Morand-Ferron J., Cole E.F., Quinn J.L. 2016 Studying the evolutionary ecology of cognition in the wild: a review of practical and conceptual challenges. Biological Reviews 91(2), 367-389. (doi:10.1111/brv.12174).

Reviewer #2: Note: I have not reviewed the original version of this manuscript submitted to PLOS ONE.

This is an interesting study on individual differences in wild baboons’ abilities to learn a two-choice color/food discrimination. I applaud the authors for their careful efforts to conduct individual cognitive tests with wild, group-living animals. Although it is not clear what the baboons had learned, the finding that even seemingly simple behavior is strongly influenced or constrained by animals’ natural environment is important. Indeed, such factors should be more frequently considered in studies of captive animals as well. However, I have some concerns and suggestions about how to effectively present and analyze the data.

INTRODUCTION

This section provides a nice exposition but the motivation for the study is not obvious to the reader. Yes, we know little about this topic in the wild, but you could make clearer why people should want to study learning (and between- and within-species variability) in the first place. There is a gap, but why do we want to fill it? What we can learn from this more broadly?

Lines 69-72: I’d add that, on the other hand, captive animals may also “fail” on problems they’d never encounter in the wild, which can distort our estimate of that species’ cognitive abilities in the other direction.

Lines 111-113: Couldn’t you argue the opposite, that males would need to learn faster because they encounter new environments? But in general, how stable are these baboons’ environments outside of seasonal variability? Even if they disperse and go to a neighboring group, aren’t they still living in the same general habitat with the same food sources?

Line 122: “because of their greater experience” I don’t fully follow this point. Greater experience with what? With that particular food item, with learning associations, something else ..?

Lines 134-135: How can they be independent and considered opposites?

Line 140-141: Why? If the idea is that neophilic individuals spend more time with the task (or would perhaps be less vigilant?) and therefore are more likely to learn, I suggest adding this explicitly – and then since you have the data, you should also test it statistically.

Line 142: Please add a brief definition for vigilance.

METHODS

The authors collected a substantive data set including food choice in the learning task and individual differences in sex, rank, neophilia, age, and vigilance in prior trials. However, several reasons make it difficult to fully appreciate this data set.

There are no indicators of reliability (except for a previous repeatability estimate for neophilia, which seems low to rely on a single measurement here).

Line 221: When trials were interrupted, did that always end the trials? If so, please state.

In lines 312-313, you say that a trial “was considered finished” when all kernels of one pile were eaten. 1) This information needs to be much earlier (move it into the Experimental Procedure section). 2) Did the experimenter remove the other pile at this point? Or was it just considered finished for data recording, but the baboon could still eat the other pile?

Line 243: “(2) the colour of the first ten kernels consumed” – should this really be 10? not “all”?

Lines 265-267: I’d add that this means all cases where they didn’t eat one pile in its entirety (20 kernels) first. It’s implied from having 20 kernels in each pile and you say this later in the manuscript, but it would be helpful for the reader here too.

Table 1: The sums by row sum to trial 1-5: 34, 37, 38, 38, 32. Why can the number of individuals for trial 1 be lower than subsequent trials? And if the grand total here is 179 and interruptions were 22, does that not contradict the descriptives in lines 400-401 and 531?

Table 1: But what is the proportion when they switched within the first 20 trials? E.g., 19 vs. 1 would still essentially be bulk feeding. The cutoff could be arbitrary, of course, but the distribution seems important because you use this criterion to subset your data.

The number of observations is extremely variable, in part due to the natural distribution of traits in this population, in part due to practical constraints during data collection, and in part due to subsetting for statistical analyses. This is not a problem per se but could be presented in a more organized fashion. The eventual reader could consult the full data set, but making it available for review as well would be extremely helpful in evaluating the analytical approach.

Line 284: You tested for multicollinearity but please add something like a correlation matrix or crosstabs to indicate which of these factors covaried. That’s interesting in its own right but also important to assess your models.

Appendix S3: How/why were these divided into low, medium, and high? Aren’t these both continuous variables?

--Models--

The analytical approach is confusing.

Couldn’t you combine Models T1, T2, …, and T5 into a single model and add trial number as a fixed effect and with interactions, like in Model T2-5? If they only learned in trial 1 and then only chose palatable trials, you’d see an interaction between trial and kernel number. If they relearned the association every time (at the same rate), you wouldn’t expect an interaction.

I also still don’t understand why you couldn’t use the full data set here too?

Including the current proportion of palatable/unpalatable kernels as a fixed effect seems odd because it’s itself dependent on prior choices, which are the measure of interest. I understand the motivation to account for this change in proportion, but this seems statistically unsound.

--Simulations--

I like this approach a lot (running simulations of the proposed processes to see what the data should look like), but this needs some work.

1. Why not also do this for between-trial learning?

2. Please provide more detail about how the simulations work. E.g., how many runs per scenario? What were the probabilities? Scenario 1: What kernel number is meant by “then” (line 362)? Scenario 2: What precisely is meant by “intermittently” (line 365) and how was it determined whether 1, 2, or 3 kernels were sampled?

3. Does “baseline probability” mean the ratio of remaining palatable/(palatable + unpalatable)? If so, “baseline” may not be the best word because it suggests the initial probability (i.e., 50%) and could therefore cause confusion.

4. Importantly, if you simulate these processes for 38 baboons, and run the same analyses that you run on the actual data (minus individual variability), can you extract these patterns? It’s not clear whether that is how you derived Table 2 (if so, state so explicitly and add the actual parameter estimates).

Given the multicollinearity concerns and criticisms of stepwise regressions, I agree with the previous Reviewer that a model selection/information theoretic approach might be more appropriate (Burnham & Anderson, 1998). It also lets you avoid overfitting, but in an arguably more principled manner (specifically, by including a penalty term for each added parameter).

RESULTS

Some descriptives (medians, ranges) for neophilia, vigilance, time spent on the task would be nice.

Lines 405-406: How many kernels did they leave uneaten in subsequent trials?

Table 3: This might change depending on changes to your analytical approach, but it’s hard to extract what probabilities/odds ratios for kernel choice these models actually predict. This would be much aided by a plot with the model fit overlaid (see also below).

FIGURES

The current figures do not convey a lot of information and only in aggregated form. And the arguably most useful figure (S4) to get a quick sense of the baboons’ behavior in the learning task is hidden in the supplemental material. The figures could be used to much greater effect to include e.g. a line + uncertainty band to indicate model fit, the number of observations (e.g., as labels or proportional to point size) or individual points/trajectories (e.g., in semi-transparent, small points or thin lines)

having Fig. S4 in the same format as Fig. 2 (simulations) would also really help comparison to the different learning processes.

Fig. 2 should have the same x- and y-axis scales throughout, to aid comparison (i.e., kernel size always from 1-5 to 36-40, baseline probability always from 0 to 1, proportion palatable always from 0 to 1).

Fig. 4 should have ranges as the labels (i.e., 1-10 instead of “10” etc.), also make the y go from 0 to 1

Fig. 4 how were low and medium determined? Also, isn’t this a continuous variable? If the categorization was just for visualization purposes, please state so in the caption.

DISCUSSION

Assuming that the general finding holds (no evidence of learning as measured by [exclusive] preference of palatable kernels, let alone individual differences), this seems fine. I appreciate the discussion of how things can go awry in the field. Two minor points to perhaps discuss in more depth:

The novel food wasn’t really novel, as the baboons had all eaten corn before. Just the color and bitter taste were new. To what extent do you think their previous experience/learning explains why they didn’t learn (more) here? If they already knew that it was nutritious and not poisonous, they may well have learned that one tasted bitter, but that might have simply not been enough to outweigh a free meal (especially when food is scarce, as you mention).

Is there any other evidence in the literature that individual differences play less of a role during times of food scarcity because everybody is similarly limited in what they can do? If so, that may be worth including.

TYPOS, WORDING SUGGESTIONS, & REFERENCES:

The use of “incorrect/correct” kernels sounds a bit awkward and seems unnecessary. I suggest simply using “unpalatable/palatable” throughout the manuscript.

Please carefully check your references. Some refs listed in the list are not listed in the text (e.g., [5, 36, 63, 64]) and refs [58] and [59] are identical.

Line 35: should be “individuals’ phenotypes” (with an s)

Line 47: should be “aspects such as foraging behavior” (add “as”)

Lines 90, 93, 95: I’d recommend italics instead of all-caps to emphasize the either/or’s

Line 110: better “female baboons would” (instead of “will”)

Line 203: should be “prior experience with corn kernels” (instead of “of”)

Line 388: should be Table 2 (instead of 1)

6. PLOS authors have the option to publish the peer review history of their article (what does this mean?). If published, this will include your full peer review and any attached files.

Reviewer #1: No

Reviewer #2: No

---

## [Author Response · Author response to Decision Letter 0]

30 Jan 2020

Response to reviewers

We would like to thank the Academic Editor and two anonymous reviewers for their helpful comments and suggestions. Our responses below are presented on a point-by-point basis (and included in a file attached). When relevant, we include the line numbers that correspond to changes made in the manuscript version without track changes. 

General comments

This has been checked 

We thank the Editor for pointing this out. Our data has been uploaded to the Figshare repository (10.6084/m9.figshare.9785867). 

3. Please include a copy of Table 2 which you refer to in your text on page 23.

A copy of Table 2 has been included in both manuscripts (track and no track). 

Additional Editor Comments

1. 39-40: please modify as follows: “current environmental challenges associated with conducting cognitive tests of animals in the wild”

This has been changed. 

2. L 494-495 “We also found that troop membership determined the likelihood of eating a higher proportion of correct kernels across trials.”: couldn’t it be done to color preference? Did you test color preference before administering the differently flavored kernels? Although I appreciated the answers you provided in reply to the comments of the previous reviewers on this issue, this aspect should be accounted for in the Discussion.

We have added the following text to the Discussion to account for the point raised by the reviewer: “Since we were avoiding any potential habituation or possible generalisation between colours, we did not test for a specific colour preference to exclude; however, as both colours used in this task represented food items frequently eaten, and thus, ecologically relevant to learn, it was not obvious at the time that the baboons would have a preference for either colour.” (L568-572)

3. L 529 “and the interaction that was contrary to prediction in trial 2”: please remind the reader what is the interaction you are mentioning here

Done. The line now reads: “the interaction between neophilia and kernel number that was contrary to prediction in trial 2”.

4. L 533: “that” repeated twice (typo)

The additional word has been removed

Reviewer #1

General comments

This study aimed to assess associative learning performance of individuals in two groups of wild baboons. The manuscript represents a significant amount of work and while it is unfortunate that the authors were not able to detect learning, I believe the trial was thoughtfully conducted. There remains a paucity of studies investigating cognition in wild populations (it is challenging!) and studies such as the current one are useful building blocks for future attempts. The authors have done a nice of addressing the concerns/ suggestions raised in the previous two reviews, as such I only have minor comments.

Minor comments

1. L51 – As there are only a small number of studies that have looked at the fitness consequences of associative learning (adaptive value) in natural environments, it would be good to cite them all here. Also, as the authors correctly point out, differences in learning are likely to reflect adaptions. The only study currently cited reports a positive correlation between learning performance and fitness correlates, but two other studies report negative relationships:

Ashton B.J., Ridley A.R., Edwards E.K., Thornton A. 2018 Cognitive performance is linked to group size and affects fitness in Australian magpies. Nature 554 (7692), 364-367 (doi:10.1038/nature25503).

Madden JR, Langley EJG, Whiteside MA, Beardsworth CE, van Horik JO (2018) The quick are the dead: pheasants that are slow to reverse a learned association survive for longer in the wild. Philosophical Transactions of the Royal Society B: Biological Sciences 373

Evans LJ, Smith KE, Raine NE (2017) Fast learning in free-foraging bumble bees is negatively correlated with lifetime resource collection. Scientific Reports 7:496

We have now added two additional references of studies that directly evaluate the fitness consequences of associative learning. Here, we also make sure to point out that there are exceptions to our statement, in particular, that differences in learning likely reflect adaptations by citing the references suggested above (L51-52). 

2. L52 – This sentence should be modified to improve clarity. I suggest inserting ‘while ultimately differences in…’ removing also from ‘individuals within a species also differ…’ and including ‘genotype or epigentic changes due to the developmental trajectory environment experienced during their lifetime.

This has been changed to: “While ultimately differences in associative learning abilities between species are likely to reflect adaptation, inter-specific differences are also likely to reflect genotype or epigenetic changes dependent on developmental trajectory, and the environment experienced during their lifetimes” (L51-55).

3. L85 – Incorrect references inserted, should be references ‘3’ (bumble bees) and ‘4’ (humingbirds). It doesn’t seem as though reference ‘5’ has been used at all – so it would pay to double check all references align as intended.

References have been checked and corrected. 

4. L86 – Rather than including a second reference for bumble bees can you reference a study that has looked at associative colour learning in foraging mammals? Or at least a different taxon e.g. butterflies, jumping spiders.

We have added new references related to studies with butterflies and birds (L86-88). 

5. Discussion – it would be good to briefly comment on baboon cognition generally. What would you expect to see in terms of their associative learning performance, based on their performance in other cognition assays, conducted either with captive or wild individuals?

We have added a few sentences that lay out our expectations in reference of baboons’ learning abilities in wild and captive conditions: “Previous studies with chacma baboons provided further support to our expectation that baboons would concisely and rapidly learn the taste-colour association. For example, captive baboons were able to solve complex learning tasks using an automated device; while wild baboons are capable of rapidly learning the location of valuable food items”. (L537-541).

6. L552 - It would be good to reinforce that studying cognition in wild populations is generally very challenging by citing the below studies:

Huebner F, Fichtel C, Kappeler PM (2018) Linking cognition with fitness in a wild primate: fitness correlates of problem-solving performance and spatial learning ability. Philosophical Transactions of the Royal Society B: Biological Sciences 373

Morand-Ferron J., Cole E.F., Quinn J.L. 2016 Studying the evolutionary ecology of cognition in the wild: a review of practical and conceptual challenges. Biological Reviews 91(2), 367-389. (doi:10.1111/brv.12174).

These references have been added (L591).

Reviewer #2

General Comments

This is an interesting study on individual differences in wild baboons’ abilities to learn a two-choice color/food discrimination. I applaud the authors for their careful efforts to conduct individual cognitive tests with wild, group-living animals. Although it is not clear what the baboons had learned, the finding that even seemingly simple behavior is strongly influenced or constrained by animals’ natural environment is important. Indeed, such factors should be more frequently considered in studies of captive animals as well. However, I have some concerns and suggestions about how to effectively present and analyze the data.

INTRODUCTION

This section provides a nice exposition but the motivation for the study is not obvious to the reader. Yes, we know little about this topic in the wild, but you could make clearer why people should want to study learning (and between- and within-species variability) in the first place. There is a gap, but why do we want to fill it? What we can learn from this more broadly?

1. Lines 69-72: I’d add that, on the other hand, captive animals may also “fail” on problems they’d never encounter in the wild, which can distort our estimate of that species’ cognitive abilities in the other direction.

We have included the reviewer’s suggestion, but we note that captive animals normally go through long period of training to habituate them to the novelty of ecologically-irrelevant tasks. The additional text reads as follows: “Conversely, the failure of captive animals to solve tasks that use unfamiliar stimuli may equally distort our estimate of that species’ cognitive abilities”. (L72-73)

2. Lines 111-113: Couldn’t you argue the opposite, that males would need to learn faster because they encounter new environments? But in general, how stable are these baboons’ environments outside of seasonal variability? Even if they disperse and go to a neighboring group, aren’t they still living in the same general habitat with the same food sources?

We agree with the reviewer and had debated this amongst ourselves. We have now deleted this justification for our prediction, and instead present only the second, stronger prediction. We did not imply that males would not have eventually learnt the association themselves, but rather that females may do so more quickly because they have a greater need to do so. 

3. Line 122: “because of their greater experience” I don’t fully follow this point. Greater experience with what? With that particular food item, with learning associations, something else ..?

We have now added: “identifying changes in food items with regards to their colour and taste” to better explain what we mean (L121-122). 

4. Lines 134-135: How can they be independent and considered opposites?

We have re-worded the sentences to clarify our meaning. The sentence now reads: “Although both responses are sometimes considered opposite ends of the same personality continuum, they are independent of one another” (L134-136).

5. Line 140-141: Why? If the idea is that neophilic individuals spend more time with the task (or would perhaps be less vigilant?) and therefore are more likely to learn, I suggest adding this explicitly – and then since you have the data, you should also test it statistically.

No, as we explain in the previous sentence, “neophilia … is associated with greater innovation and successful problem-solving as individuals are more likely to explore a novel situation”. To clarify, we have now changed the wording to: “Based on these previous findings, …”(L140-141)

6. Line 142: Please add a brief definition for vigilance.

We have added a definition (L142-143). 

METHODS

The authors collected a substantive data set including food choice in the learning task and individual differences in sex, rank, neophilia, age, and vigilance in prior trials. However, several reasons make it difficult to fully appreciate this data set.

7. There are no indicators of reliability (except for a previous repeatability estimate for neophilia, which seems low to rely on a single measurement here).

The repeatability of neophilia is in line with average repeatability for behaviour (Bell et al., 2009), and single measurements were made to decrease the number of times the baboons were presented with stimuli from the observers. This single within-season measurement appears to be biologically meaningful, predicting, for example, the way individuals use personal and social information during foraging (e.g. Carter et al. 2014). Regarding other behaviours that were quantified from the videos, each video was processed by one of the authors (CML) at least three times to quantify kernel choice, number of kernels and number of vigilance events. In cases when the measurement varied from one coding to the next, we re-coded the behaviour once more. We consider these measurements to be reliable, as there was little ambiguity. To address this, we now included a small paragraph in the “Experimental protocol” section, defining sources of potential ambiguity and noting the steps taken to address such irregularities.

Bell, Alison M., Shala J. Hankison, and Kate L. Laskowski. "The repeatability of behaviour: a meta-analysis." Animal behaviour 77.4 (2009): 771-783.

Carter, Alecia J., et al. "Personality predicts the propensity for social learning in a wild primate." PeerJ 2 (2014): e283.

8. Line 221: When trials were interrupted, did that always end the trials? If so, please state.

We have added this information (L240-242). 

9. In lines 312-313, you say that a trial “was considered finished” when all kernels of one pile were eaten. 1) This information needs to be much earlier (move it into the Experimental Procedure section). 2) Did the experimenter remove the other pile at this point? Or was it just considered finished for data recording, but the baboon could still eat the other pile?

This information was added in the Statistical Analysis section as it relates to how we selected the data under analysis, that is, that in a given trial, all observations after the first pile was consumed entirely were not considered for the given reasons. To clarify this, we have changed the wording to: “we did not consider individuals’ choices after all the kernels of one pile were eaten” (L331-332). 

We were unable to remove the remaining pile of kernels while the trial was ongoing for risk of aggression from the baboons. 

10. Line 243: “(2) the colour of the first ten kernels consumed” – should this really be 10? not “all”?

We thank the Reviewer for pointing this out. We have changed accordingly.

11. Lines 265-267: I’d add that this means all cases where they didn’t eat one pile in its entirety (20 kernels) first. It’s implied from having 20 kernels in each pile and you say this later in the manuscript, but it would be helpful for the reader here too.

We have added this information (L291).

12. Table 1: The sums by row sum to trial 1-5: 34, 37, 38, 38, 32. Why can the number of individuals for trial 1 be lower than subsequent trials? And if the grand total here is 179 and interruptions were 22, does that not contradict the descriptives in lines 400-401 and 531?

Table 1: But what is the proportion when they switched within the first 20 trials? E.g., 19 vs. 1 would still essentially be bulk feeding. The cutoff could be arbitrary, of course, but the distribution seems important because you use this criterion to subset your data.

Although we tested 37 animals with a first trial, we were unable to extract data from three recordings, as the camera malfunctioned in these cases, hence the difference in sample size between trial 1 and the rest of the trials. We have a total of 162 trials, out of which we were able to extract data from 159 (the total number of trials on the table). The interruptions listed on the table represent interruptions that happened before animals could sample both piles. We have added a line to the results to specify the difference between overall interruptions and interruptions which happened before both piles of kernels were samples (L421-423). The number of interruptions described in L531 corresponds to the interruptions that happened after both piles were sampled, but which nonetheless affected the patterns under analysis. In Table 1, we included all individuals who had sampled both colours before consuming one pile of kernels entirely, regardless of the proportion when they switched, as we believed this offered more consistency; while the cutoff was decided only when a single pile of kernels remained. 

13. The number of observations is extremely variable, in part due to the natural distribution of traits in this population, in part due to practical constraints during data collection, and in part due to subsetting for statistical analyses. This is not a problem per se but could be presented in a more organized fashion. The eventual reader could consult the full data set, but making it available for review as well would be extremely helpful in evaluating the analytical approach.

Our dataset is available in an open repository and we have provided the DOI with this resubmission to access it. 

14. Line 284: You tested for multicollinearity but please add something like a correlation matrix or crosstabs to indicate which of these factors covaried. That’s interesting in its own right but also important to assess your models.

We have added a covariance table in the supplementary material section as requested. 

15. Appendix S3: How/why were these divided into low, medium, and high? Aren’t these both continuous variables?

Yes, these variables were continuous, but for the purposes of ease-of-assessment for the reader, in this table each variable was separated evenly into three categories according to tertiles (bottom third, middle third, high third). We have now added more detail in our description of the Appendix ( L846-849).

--Models--

The analytical approach is confusing.

Couldn’t you combine Models T1, T2, …, and T5 into a single model and add trial number as a fixed effect and with interactions, like in Model T2-5? If they only learned in trial 1 and then only chose palatable trials, you’d see an interaction between trial and kernel number. If they relearned the association every time (at the same rate), you wouldn’t expect an interaction.

I also still don’t understand why you couldn’t use the full data set here too?

We had initially combined all models and as the reviewer suggests and had added trial number as a fixed effect. However, this approach was difficult to interpret and gave us very limited information about differences in animals’ learning within each trial. To understand whether and how within-trial learning occurred, we would have to assess each model using post-hoc tests in the manner we have now done. Consequently, we decided to evaluate each trial separately. Moreover, since the simulations we proposed predicted several possible patterns, a single model would have failed to capture this. 

Including the current proportion of palatable/unpalatable kernels as a fixed effect seems odd because it’s itself dependent on prior choices, which are the measure of interest. I understand the motivation to account for this change in proportion, but this seems statistically unsound.

We have given this quite some thought. To reiterate, our rationale for using this variable was to account for the palatable kernels still available as a trial progressed, since individuals would be more likely to choose randomly an unpalatable kernel as the number of the palatable kernels decreased and vice versa. Not controlling for this could lead to spurious conclusions e.g. that the baboons showed learning as the trial progressed, but that this was just a function of randomly choosing many unpalatable kernels to begin with, before having few unpalatable options to choose and thus making “correct” choices as the trial progressed. We understand the reviewer’s concern, but we believe that it is important to control statistically for the change in the underlying probability of randomly choosing a correct kernel. 

--Simulations--

I like this approach a lot (running simulations of the proposed processes to see what the data should look like), but this needs some work.

1. Why not also do this for between-trial learning?

The predictions for between-trial learning were clearer than those for within-trial learning, in large part because (a) this is the more standard approach in this field and (b) bulk feeding was not an issue for between-trial learning (as we did not interrupt trials after the first pile had been eaten).

2. Please provide more detail about how the simulations work. E.g., how many runs per scenario? What were the probabilities? Scenario 1: What kernel number is meant by “then” (line 362)? Scenario 2: What precisely is meant by “intermittently” (line 365) and how was it determined whether 1, 2, or 3 kernels were sampled?

The simulated scenarios were developed post hoc based on our observations of the feeding patterns seen throughout trials, in particular, the bulk feeding. The simulations were used to generate patterns with which to compare our observed data. As such, these are not iterative statistical simulations. 

We have made some additions in the text to clarify this, and now refer to “scenarios” c.f. “simulations”. In particular, we have added the following text: “Scenarios were not probabilistic and as such there is only one outcome for each scenario, as described.” (L390-391).

3. Does “baseline probability” mean the ratio of remaining palatable/(palatable + unpalatable)? If so, “baseline” may not be the best word because it suggests the initial probability (i.e., 50%) and could therefore cause confusion.

Yes, the baseline probability referred to the probability of choosing a correct kernel given the numbers of kernels that were available for each choice. To avoid any confusion, we have removed the term from the manuscript.

4. Importantly, if you simulate these processes for 38 baboons, and run the same analyses that you run on the actual data (minus individual variability), can you extract these patterns? It’s not clear whether that is how you derived Table 2 (if so, state so explicitly and add the actual parameter estimates).

As mentioned above, we did not conduct statistical, iterative simulations, but rather developed each scenario based on our observations of the feeding patters of the baboons. We hope the new edits make this clearer.

Given the multicollinearity concerns and criticisms of stepwise regressions, I agree with the previous Reviewer that a model selection/information theoretic approach might be more appropriate (Burnham & Anderson, 1998). It also lets you avoid overfitting, but in an arguably more principled manner (specifically, by including a penalty term for each added parameter).

Regarding the concerns the reviewer highlights, we controlled for multicollinearity through VIFs as suggested by Zuur et al. [1] and have provided a correlation matrix showing Spearman rank correlations between all fixed effects variables.

Regarding the criticisms of stepwise model selection, disciplined hypothesis testing for small amounts of model reduction, which we do here, is considered as appropriate [2]. Given that we controlled for both issues the reviewer mentions, and that post-analysis changes to modelling approaches are also criticised, we are reticent to change our analysis at this stage. 

RESULTS

1. Some descriptives (medians, ranges) for neophilia, vigilance, time spent on the task would be nice.

The median and range for neophilic behaviour has been added in L175. Median and range for vigilance and time spent on the task were added on L435-436 and L424 respectively.

2. Lines 405-406: How many kernels did they leave uneaten in subsequent trials?

Since we were comparing the pilot study to the current study, we focused only on the first trial, as any kernels left uneaten in subsequent trials would have been left on account of potential learning from previous trials and would not have been a fair comparison to the pilot study. 

Table 3: This might change depending on changes to your analytical approach, but it’s hard to extract what probabilities/odds r atios for kernel choice these models actually predict. This would be much aided by a plot with the model fit overlaid (see also below).

We have included in Table 3 a conversion of the estimate (in logit scale) to probability. We hope this offers further information and clarity on our models. We have also modified our figures to incorporate the reviewer’s suggestions (see our answer to comment 1 FIGURES below). 

FIGURES

1. The current figures do not convey a lot of information and only in aggregated form. And the arguably most useful figure (S4) to get a quick sense of the baboons’ behavior in the learning task is hidden in the supplemental material. The figures could be used to much greater effect to include e.g. a line + uncertainty band to indicate model fit, the number of observations (e.g., as labels or proportional to point size) or individual points/trajectories (e.g., in semi-transparent, small points or thin lines)

having Fig. S4 in the same format as Fig. 2 (simulations) would also really help comparison to the different learning processes.

We have added the figure previously depicted on S4 to the main text (now Fig 3) as the reviewer suggested and have added the number of observations as labels. We have additionally modified the other two figures (Fig 4 & 5) to include individual points.

2. Fig. 2 should have the same x- and y-axis scales throughout, to aid com0parison (i.e., kernel size always from 1-5 to 36-40, baseline probability always from 0 to 1, proportion palatable always from 0 to 1).

We had originally plotted the graphs as the reviewer has suggested. However, these plots are merely visual aids of what the patterns of consumption of kernels would look like given our within-trial learning scenarios. Altering the x-axis to include all kernels (i.e. 1-5 to 36-40), affected the visualization of the predicted effect of each scenario, particularly in scenarios 3 and 5, where changing the scales of both axes severely impacted the visualization and interpretation of the plots. We ultimately decided to change this to enable a clearer visualization of the possible trends. 

3. Fig. 4 should have ranges as the labels (i.e., 1-10 instead of “10” etc.), also make the y go from 0 to 1

This has been done.

4. Fig. 4 how were low and medium determined? Also, isn’t this a continuous variable? If the categorization was just for visualization purposes, please state so in the caption.

This information has now been added to the legend of the figure. 

DISCUSSION

Assuming that the general finding holds (no evidence of learning as measured by [exclusive] preference of palatable kernels, let alone individual differences), this seems fine. I appreciate the discussion of how things can go awry in the field. Two minor points to perhaps discuss in more depth:

1. The novel food wasn’t really novel, as the baboons had all eaten corn before. Just the color and bitter taste were new. To what extent do you think their previous experience/learning explains why they didn’t learn (more) here? If they already knew that it was nutritious and not poisonous, they may well have learned that one tasted bitter, but that might have simply not been enough to outweigh a free meal (especially when food is scarce, as you mention).

In the “Experimental Procedure”, we clarify that: “we considered it unlikely that these individuals’ responses to the task would be any different, as all the baboons had prior experience with corn kernels [3] and none had previously participated in a task involving colour cues indicating variation in palatability” (L204-207). We also detail in the Discussion section the possibility that animals learn that one option was “safe”, albeit bitter tasting, and that ultimately the kernels were too nutritious to avoid entirely (L550-552). To this end, we have now included a few of lines to reflect your suggestion (L557-559). 

2. Is there any other evidence in the literature that individual differences play less of a role during times of food scarcity because everybody is similarly limited in what they can do? If so, that may be worth including.

While we were unable to find a study that directly tested the role of individual differences in response to the availability of food resources, we wanted to reflect the reviewer’s comment in our Discussion. We have now added the text: “Despite substantial literature detailing the influence of unfavourable environment on animals’ cognitive abilities and cue use, there is little empirical evidence on the phenotypic plasticity of labile traits related to cognition. Consequently, we have a poor understanding of cognitive variability in fluctuating, rather than predictable and constant, environments” (L597-601)

Minor Comments 

1. The use of “incorrect/correct” kernels sounds a bit awkward and seems unnecessary. I suggest simply using “unpalatable/palatable” throughout the manuscript.

We have made these changes throughout. 

2. Please carefully check your references. Some refs listed in the list are not listed in the text (e.g., [5, 36, 63, 64]) and refs [58] and [59] are identical.

References have been checked. 

3. Line 35: should be “individuals’ phenotypes” (with an s)

Done.

4. Line 47: should be “aspects such as foraging behavior” (add “as”)

Done.

5. Lines 90, 93, 95: I’d recommend italics instead of all-caps to emphasize the either/or’s

Done. 

6. Line 110: better “female baboons would” (instead of “will”)

Done. 

7. Line 203: should be “prior experience with corn kernels” (instead of “of”)

Done. 

8. Line 388: should be Table 2 (instead of 1)

Done.

References

1. Zuur AF, Ieno EN, Walker N, Saveliev AA, Smith GM. Mixed Effects Models And Extensions In Ecology With R [Internet]. Springer, editor. New York, NY: Springer New York; 2009. (Statistics for Biology and Health). 

2. Bolker BM, Brooks ME, Clark CJ, Geange SW, Poulsen JR, Stevens MHH, et al. Generalized Linear Mixed Models: A Practical Guide For Ecology And Evolution. Trends Ecol Evol. 2009;24(3):127–35. 

3. Marshall HH, Carter AJ, Ashford A, Rowcliffe JM, Cowlishaw G. Social Effects On Foraging Behavior And Success Depend On Local Environmental Conditions. Ecol Evol. 2015;5(2):475–92.

---

## [Decision Letter · Decision Letter 1]

4 Mar 2020

PONE-D-19-25653R1

Exploring individual variation in associative learning abilities through an operant conditioning task in wild baboons

PLOS ONE

Dear Dr. Carter,

Thank you for submitting your manuscript to PLOS ONE. After careful consideration, we feel that it has merit but does not fully meet PLOS ONE’s publication criteria as it currently stands. Therefore, we invite you to submit a revised version of the manuscript that addresses the points raised during the review process.

I require to address the following very minor points before the manuscript can be accepted: 

- L 40: two full stops (typo)

- L72-73: the sentence is a bit unclear, I suggest to modify it as follows: “Conversely, the failure of captive animals to solve tasks that they would never encounter in the wild may equally distort our estimate of that species’ cognitive abilities”.

- L 83: two square brackets (typo)

- L 135: I suggest to delete the word “personality” here, since personality is a complex construct; similarly, on l 480 I suggest to replace “personality” with “neophilia”

- L 323: the parenthesis at the end of the sentence is missing

- L 422, 540 please replace the semicolon with a comma

- L 544 (and 561) since “tendency” has a very precise meaning in statistics (0.05 < p < 0.1), unless this is the case, I suggest to replace this word with an alternative one

- L 551 please erase the comma after “study”

We would appreciate receiving your revised manuscript by Apr 18 2020 11:59PM. To enhance the reproducibility of your results, we recommend that if applicable you deposit your laboratory protocols in protocols.io, where a protocol can be assigned its own identifier (DOI) such that it can be cited independently in the future. For instructions see: http://journals.plos.org/plosone/s/submission-guidelines#loc-laboratory-protocols

We look forward to receiving your revised manuscript.

Kind regards,

Elsa Addessi

Academic Editor

PLOS ONE

Reviewers' comments:

Reviewer's Responses to Questions

**Comments to the Author**

1. If the authors have adequately addressed your comments raised in a previous round of review and you feel that this manuscript is now acceptable for publication, you may indicate that here to bypass the “Comments to the Author” section, enter your conflict of interest statement in the “Confidential to Editor” section, and submit your "Accept" recommendation.

Reviewer #1: All comments have been addressed

Reviewer #2: All comments have been addressed

2. Is the manuscript technically sound, and do the data support the conclusions?

Reviewer #1: Yes

Reviewer #2: Yes

3. Has the statistical analysis been performed appropriately and rigorously? 

Reviewer #1: Yes

Reviewer #2: Yes

4. Have the authors made all data underlying the findings in their manuscript fully available?

Reviewer #1: Yes

Reviewer #2: Yes

5. Is the manuscript presented in an intelligible fashion and written in standard English?

Reviewer #1: Yes

Reviewer #2: Yes

6. Review Comments to the Author

Reviewer #1: All minor corrections requested have been suitably addressed by authors. I look forward to seeing this work published.

Reviewer #2: This paper details a study on individual differences in wild baboons’ abilities to learn a two-choice color/food discrimination. It nicely highlights that even well-designed and seemingly simple tests of animal behavior and cognition can be heavily affected by environmental conditions. I thank the authors for their clarifications in the manuscript and thoughtful responses to the comments/suggestions raised in the review process.

7. PLOS authors have the option to publish the peer review history of their article (what does this mean?). If published, this will include your full peer review and any attached files.

Reviewer #1: No

Reviewer #2: No

---

## [Author Response · Author response to Decision Letter 1]

6 Mar 2020

1. L 40: two full stops (typo)

This has been corrected. 

2. L72-73: the sentence is a bit unclear, I suggest to modify it as follows: “Conversely, the failure of captive animals to solve tasks that they would never encounter in the wild may equally distort our estimate of that species’ cognitive abilities”.

This sentence has been modified as suggested. 

3. L 83: two square brackets (typo)

The additional bracket has been removed.

4. L 135: I suggest to delete the word “personality” here, since personality is a complex construct; similarly, on l 480 I suggest to replace “personality” with “neophilia”

All mentions of “personality” have been replaced by the work “neophilia”.

5. L 323: the parenthesis at the end of the sentence is missing

This has been added. 

6. L 422, 540 please replace the semicolon with a comma

Done.

7. L 544 (and 561) since “tendency” has a very precise meaning in statistics (0.05 < p < 0.1), unless this is the case, I suggest to replace this word with an alternative one.

We have modified each sentence to replace the word “tendency”. In L 544-545 the sentence now reads: “This interpretation is supported by our observation that the baboons left slightly more bitter kernels uneaten”. In L562-563 the sentence now reads: “Such a result may be indicative of a species’ selective attention towards red food items”.

8. L 551 please erase the comma after “study”

Done.

---

## [Editor Report · Decision Letter 2]

10 Mar 2020

Exploring individual variation in associative learning abilities through an operant conditioning task in wild baboons

PONE-D-19-25653R2

Dear Dr. Carter,

We are pleased to inform you that your manuscript has been judged scientifically suitable for publication and will be formally accepted for publication once it complies with all outstanding technical requirements.

With kind regards,

Elsa Addessi

Academic Editor

PLOS ONE
---

## [Editor Report · Acceptance letter]

13 Mar 2020

PONE-D-19-25653R2 

Exploring individual variation in associative learning abilities through an operant conditioning task in wild baboons 

Dear Dr. Carter:

I am pleased to inform you that your manuscript has been deemed suitable for publication in PLOS ONE. Congratulations! Your manuscript is now with our production department. 

With kind regards,

on behalf of

Dr. Elsa Addessi 

Academic Editor

PLOS ONE